# Comprehensive and Rapid Quality Evaluation Method for the Ayurvedic Medicine Divya-Swasari-Vati Using Two Analytical Techniques: UPLC/QToF MS and HPLC–DAD

**DOI:** 10.3390/ph14040297

**Published:** 2021-03-27

**Authors:** Acharya Balkrishna, Sudeep Verma, Priyanka Sharma, Meenu Tomer, Jyotish Srivastava, Anurag Varshney

**Affiliations:** 1Drug Discovery and Development Division, Patanjali Research Institute, Haridwar 249 405, Uttarakhand, India; acharya.balkrishnapri@prft.in (A.B.); sudeep.verma@prft.in (S.V.); priyanka.sharma@prft.in (P.S.); meenu.tomer@prft.in (M.T.); jyotish.srivastava@prft.in (J.S.); 2Department of Allied and Applied Sciences, University of Patanjali, Haridwar 249 405, Uttarakhand, India

**Keywords:** Ayurveda, Divya-Swasari-Vati, herbal medicine, UPLC/QToF MS, HPLC, validation

## Abstract

Divya-Swasari-Vati (DSV) is a calcium-containing herbal medicine formulated for the symptomatic control of respiratory illnesses observed in the current COVID-19 pandemic. DSV is an Ayurvedic medicine used for the treatment of chronic cough and inflammation. The formulation has shown its pharmacological effects against SARS-CoV-2 induced inflammation in the humanized zebrafish model. The present inventive research aimed to establish comprehensive quality parameters of the DSV formulation using validated chromatographic analytical tools. Exhaustive identification of signature marker compounds present in the plant ingredients was carried out using ultra performance liquid chromatography-quadrupole time-of-flight mass spectrometry (UPLC/QToF MS). This was followed by simultaneous estimation of selected marker components using rapid and reliable high-performance liquid chromatography (HPLC) analysis. Eleven marker components, namely gallic acid, protocatechuic acid, methyl gallate, ellagic acid, coumarin, cinnamic acid, glycyrrhizin, eugenol, 6-gingerol, piperine and glabridin, were selected out of seventy-four identified makers for the quantitative analysis in DSV formulation. Validation of the HPLC method was evaluated by its linearity, precision, and accuracy tests as per the International Council of Harmonization (ICH) guidelines. Calibration curves for the eleven marker compounds showed good linear regression (*r*^2^ > 0.999). The relative standard deviation (RSD) value of intraday and interday precision tests were within the prescribed limits. The accuracy test results ranged from 92.75% to 100.13%. Thus, the present inclusive approach is first of its kind employing multi-chromatographic platforms for identification and quantification of the marker components in DSV, which could be applied for routine standardization of DSV and other related formulations.

## 1. Introduction

The world community is grappling with the devastating effects of the novel coronavirus disease (COVID-19) caused by Severe Acute Respiratory Syndrome Corona Virus 2 (SARS-CoV-2). The pandemic has caused a serious medical crisis, infecting more than 120 million people and leading to more than 2 million deaths [1]. The situation is considered to be more serious for patients suffering from respiratory syndromes. Infection with this respiratory virus is associated with robust inflammatory responses, which further worsen the condition [2]. The immune system plays an essential role in COVID-19 infection. Hence, enhancing the (natural body system) immunity may represent a major contribution as a prophylactic measure against multiple pathogenic conditions as well as maintaining optimum health [3].

Currently, the pandemic has entered a perilous phase where there are no specific drugs or other therapeutics against this viral outbreak [4]. The scientific community is working relentlessly to discover active pharmacological moieties that might provide new tools against this unabated transmission. Traditional, complementary and alternative medicines have emerged as the bright ray of hope in this regard [5]. Since immune dysfunction plays a vital role in disease progression, consumption of herbal medicines containing certain active compounds which have antimicrobial or antiviral, anti-inflammatory and immuno-stimulatory activities, might have potentials as effective prophylactic or even therapeutic against SARS-CoV-2 [6].

Divya-Swasari-Vati (DSV) is a calcio-herbal tablet formulation consisting of sixteen herbo-mineral ingredients (Table 1). The formulation is concocted using different parts of several medicinal plants which have a long history of usage for the treatment of respiratory infections and bronchitis. Herbal ingredients like roots of *Glycyrrhiza glabra* (licorice) have been used ethno-medicinally for the treatment of coughs, cold and COPD. Glycyrrhizin, a triterpenoid saponin from licorice has performed remarkably in inhibiting the replication of earlier SARS virus with very few side effects [7]. Eugenol, one of the abundant phenolics found in the buds of *Syzygium aromaticum* and bark of *Cinnamomum zeylanicum* (cinnamon), is very well known for its anti-inflammatory and free radical scavenging properties [8]. *Pistacia integerrima* (zebrawood) is known to exert anti-asthmatic action by mitigating TNFα activity [9]. *Cressa cretica* is known to have bronchodilatory and mast cell-stabilizing activity [10]. *Zingiber officinale* (ginger), has been used for ages as a home remedy for the treatment of common cold, asthma and bronchitis. A novel compound having structural similarities with 6-gingerol showed strong binding affinities SARS-CoV-2 viral receptors [11]. Piperine from the fruits of *Piper nigrum* (black pepper) and *Piper longum* (long pepper), has been shown to possess endothelial barrier protective and leukocyte migration suppressive effects [12]. Secondary metabolites from the roots of *Anacyclus pyrethrum* (Spanish chamomile) like saponins and tannins are known to exert immunomodulatory and immune-stimulating effects [13]. The ethno-medicinal uses of DSV ingredients have been recently validated in a mouse model of allergic asthma where the ingredients potentially suppressed the allergic asthma by modulating pro-inflammatory cytokines [14]. It is well established that the pathophysiology of SARS-CoV-2 infectivity involves different pro-inflammatory cytokines, which put the host immune system into overdrive. Thus, blocking the cytokine storm could represent a vital weapon for combating SARS-CoV-2 infectivity. Indeed, DSV successfully ameliorated SARS-CoV-2 spike protein-induced inflammation in a humanized zebrafish model by blocking the IL-6 and TNF α cytokine surge [15].

Plant extracts are exceedingly complex multicomponent mixtures. These wide arrays of phytochemical components may either function alone or in amalgamation with other components to yield the desired pharmacological effects [16]. Chromatographic fingerprinting and chemical profiling are very much essential for global acceptance of traditional herbal medicines (THMs); and have proved to be a favorable approach to ensure quality control of herbal preparations. Many agencies such as the World Health Organization (WHO), the Food and Drug Administration (FDA), and the European Medicines Agency (EMA) recommend the use of analytical modern analytical tools to monitor critical quality attributes of in-process materials in a timely manner. This approach is quintessential to verify the stability and consistency of THMs [17,18]. Poly-herbs of DSV consist of a myriad of secondary metabolites. Consequently, in order to standardize the formulation, and to help manufacturers to have consistent products, a suitable selection of analytical techniques becomes imperative.

Thus, for the comprehensive quality control of DSV, we describe herein the development of a simple, reliable, and sensitive high-performance liquid chromatography–diode array detection (HPLC–DAD) method for the simultaneous analysis of eleven marker components in the formulation. The intrinsic complexity of THMs with no obvious targets for quantification is one of the biggest challenges when it comes to ensuring their identity and quality. Ultra-performance liquid chromatography–mass spectrometry coupled with a quadrupole time of flight analyzer (UPLC/QToF MS) is one of the most powerful analytical tools which excels in the identification of ionisable moieties with high mass accuracy [19]. Seventy-four compounds were characterized in the DSV formulation using UPLC/QToF MS out of which eleven—gallic acid, protocatechuic acid, methyl gallate, ellagic acid, coumarin, cinnamic acid, glycyrrhizin, eugenol, 6-gingerol, piperine and glabridin—were chosen as the signature analytes of the formulation. A validated HPLC method was then successfully applied for the simultaneous quantification of target components in five different batches of DSV.

## 2. Results

### 2.1. UPLC/QToF MS Analysis Characterized Chemical Markers in DSV

Peaks corresponding to chemical metabolites in DSV (Figure 1) were identified using the UPLC/QToF MS system and have been listed in Table 2. Fifty-nine compounds were identified in the positive mode of ionization (Figure 1A, Table 2) and forty-five compounds were identified in the negative mode of ionization (Figure 1B, Table 2). Thirty common compounds were found in both the ionization modes, i.e., positive and negative modes. Eleven markers (gallic acid, protocatechuic acid, methyl gallate, ellagic acid, coumarin, cinnamic acid, glycyrrhizin, eugenol, 6-gingerol, piperine and glabridin, Appendix A) were selected out of seventy-four identified compounds as chemical markers to represent the herbal components in the DSV formulation. The identification of compounds relied on the mass fragmentation pattern data and accurate mass measurement of the selected chemical markers with the aid of a mass spectral library created in-house and reported literature values (Appendix A). The triterpenoid glycyrrhizin and the isoflavonnoid glabridin were selected as the signature markers for *Glycyrrhiza glabra*. Eugenol, a phenylpropanoid derivative, and cinnamic acid were chosen for *Sygygium aromaticum* and *Cinnamomum zelanicum*, respectively. Methyl gallate, one of the active constituents present in the galls of *Pistacia integerrima*, was selected as its signature marker. Coumarins are the biologically active constituents of the halophytic plant *Cressa cretica* hence coumarin was selected as the marker for that species. 6-Gingerol, a very well-known pungent phenol from *Zingiber officinale* was designated as the marker for this plant. The alkaloid piperine was chosen as the representative marker for *Piper nigrum* and *Piper longum*. Roots of *Anacylus pyrethrum* are rich in tannins, hence, the most popular tannins—gallic acid and ellagic acid—were selected for the same.

### 2.2. Establishment and Optimization of the HPLC–DAD Method:

Chromatographic separation seems to be a challenging task when it comes to structurally diversified phyto-components for these compounds possess very broad range of polarity. The aim was to separate the targeted components gallic acid, protocatechuic acid, methyl gallate, ellagic acid, coumarin, cinnamic acid, glycyrrhizin, eugenol, 6-gingerol, piperine and glabridin with a compatible solvent system. Compared with isocratic elution, gradient elution gave a shorter overall analysis and optimum resolution. After several trials, the best separation of all the marker components was found with a solvent system consisting 0.1% orthophosphoric acid in water adjusted to pH 2.5 with diethylamine (solvent A) and 0.1% orthophosphoric acid in acetonitrile: water (88:12) adjusted to pH 2.5 (solvent B) with gradient programming. Finally, optimized chromatographic conditions to ensure good separation were achieved by injecting 10 μL of standard and sample solution using a Shodex C18-4E (5 µm, 4.6 mm × 250 mm) maintained at 35 °C and subjected to binary gradient elution. The wavelengths at which all the signature analytes were detected were found to be 278 nm and 250 nm. The chromatograms, acquired with a flow rate of 1.0 mL/min showed effective separation of analytes (Figure 2).

### 2.3. Validation of the Developed and Optimized HPLC Method for Quantitative Analysis of Eleven Marker components in DSV

The HPLC method was validated by defining the linearity, limits of quantification and detection, accuracy, precision, robustness and ruggedness. Validation was performed on DSV (batch #B SWV117) of as per the requirements established by ICH guidelines [20].

#### 2.3.1. Specificity, Linearity, Limits of Quantification and Detection

No interference was detected close to the retention times of the selected marker components indicating that the detected peaks were free from co-eluting interferents. The result indicates that the peak of the analyte was pure which confirmed the specificity of the method (Appendix A). The linear regression analysis data for the calibration plot exhibited good linear relationship for all the compounds over the concentration range proposed. The correlation coefficient for the calibration curves of all the targeted signature analytes was found to be higher than 0.99 (Appendix A). The results of regression equation, the correlation coefficient (*r*^2^) along with the concentration range are listed in Table 3. The LOD of marker components was found to below the prescribed limit (NMT 33%) whereas, the LOQ values were also within the assigned permissible range (NMT 10%) (Table 3).

#### 2.3.2. Accuracy and Precision

The recoveries of the eleven marker compounds at the three different concentrations were observed to be in the range from 92.75% to 100.13%. The results provided evidence that the established HPLC method is accurate for the simultaneous determination of eleven marker components in DSV (Table 3). Precision in interday and intraday runs are shown in Table 3. The values of the precision were within the permissible criteria of ˂2% for gallic acid, protocatechuic acid, methyl gallate, ellagic acid, coumarin, cinnamic acid, glycyrrhizin, eugenol, 6-gingerol, piperine and glabridin indicating that the method is sufficiently precise for them (Table 3).

#### 2.3.3. Robustness and Ruggedness

Deliberate variations in terms of column temperature and flow rate were taken into consideration for this method (Table 3). In all modifications, good separation of targeted analytes was achieved, and it was observed that %RSD was within the limit of not more than 20% indicating the robustness of the method. All the marker components showed %RSD less than 10% except methyl gallate which was found to be 15.63%. Ruggedness for the developed HPLC method was calculated by the %RSD of intermediate precision. The results in Table 3 show that % RSD for gallic acid, protocatechuic acid, methyl gallate, ellagic acid, coumarin, cinnamic acid, glycyrrhizin, eugenol, 6-gingerol, piperine and glabridin (NMT 10%) which indicates the ruggedness of the developed method for the analysis of the targeted analytes.

### 2.4. Validated HPLC–DAD Method Simultaneously Quantified Eleven Marker Analytes in Five Different Batches of DSV

The validated method was applied for the simultaneous determination of eleven marker components in five batches of DSV. The results of quantitative analysis are depicted in Figure 3. It was viewed, that the signature analytes, gallic acid, glycyrrhizin, eugenol and piperine showed marked prominence in all the batches of formulation. It is obvious from the results that detection of a single component cannot control the quality of DSV effectively. Thus simultaneous determination of multiple markers becomes imperative in this situation. Our developed HPLC method suitably detected the targeted analytes in all five DSV batches, with acceptable batch to batch variance. Gallic acid (3438 ± 550.7 µg/g), glycyrrhizin (4214 ± 201.9 µg/g), eugenol (5385 ± 980.2 µg/g) and piperine (5763 ± 699.4 µg/g) stood out in terms of showing marked prominence in DSV batches, whereas, the mean concentration of—protocatechuic acid, methyl gallate, ellagic acid, coumarin, cinnamic acid, 6-gingerol and glabridin—were found to be 65.79 ± 9.109 µg/g, 875.7 ± 260.3 µg/g, 283.3 ± 68.82 µg/g, 49.85 ± 8.979 µg/g, 40.24 ± 2.514 µg/g, 494.1 ± 34.03 µg/g, and 241.2 ± 39.32 µg/g, respectively (Figure 3).

## 3. Discussion

For millennia, traditional herbal medicines (THMs) have proven their value as sources of active therapeutic molecules [21]. THMs are made up of multiple herbs containing a plethora of secondary metabolites in variable concentrations. Phyto-therapeutics are complex, systematic and multi-targeted which are claimed to work synergistically [22]. The quality of THM products are usually influenced by their different plant species, growing conditions, harvest seasons, processing and other factors, which [23] have made their use more challenging. A number of attempts have been made in the academic and industrial settings, for mitigating the attrition rates of herbal drug development and their translatability to human applications. The intrinsic complexities associated with the botanicals demand the development of novel analytical procedures for reviving their lost translational capabilities [24]. The development and validation of analytical procedures plays a pivotal role in discovery, development, and manufacture of pharmaceuticals [25]. Validated test procedures further verify that the proposed analytical method is accurate and reliable for the assessment of APIs in a given drug preparation [26].

Examination of complex herbal blends bears several essential issues and significant challenges. Subsequently the identification and quantification of desired chemical markers becomes imperative, which further ensures their safety and efficacy [27]. Marker-based standardization of medicinal plants is a widely accepted and reliable technique. Ideally, the markers are selected on the basis of their therapeutic action (active constituent marker). These components must be stable and most importantly must be present consistently in the ingredients as well as in their respective formulations [28]. Another relevant criterion for their selection relies on the ease of isolation, characterization and availability. In this study, ultra-liquid chromatography coupled to quadruple time of flight (QToF) mass spectrometry was used for identification and selection of analytical markers for quality control of DSV formulation. The technique offers very high resolution and selectivity in terms of providing abundant mass information, with accurate mass measurements, and, therefore is quite useful for identifying the target compounds thoroughly [29]. Based on the existing literature [30] and the analyst’s own expertise, a UPLC/QToF/MS method was developed. The developed method was utilized to identify seventy-four (74) phyto-metabolites in the DSV formulation. For example, compound number **1**, showed *m/z* 191.0555 in negative ionization mode, its respective mass fragmentation pattern was observed to be *m/z* 173.0445, *m/z* 149.0443, *m/z* 129.0184, *m/z* 113.0258, *m/z* 89.0267 which confirmed the presence of quinic acid (192.0634 Da) with [H^-^] adduct. Compound number **3** was detected in negative ionization mode and showed *m/z* 169.0136, so the compound was confirmed as gallic acid (170.0215 Da), by its mass fragmentation pattern in which peaks were observed at *m/z* 153.0177, *m/z* 137.0238, *m/z* 125.0238 with [H^-^] adduct. Likewise, seventy four compounds were identified and confirmed in the formulation on the basis of their accurate mass screening and fragmentation patterns as depicted in Table 2. Appendix A. Eleven markers—gallic acid, protocatechuic acid, methyl gallate, ellagic acid, coumarin, cinnamic acid, glycyrrhizin, eugenol, 6-gingerol, piperine and glabridin— were selected out of seventy four identified compounds. The strategies behind the selection of the targeted eleven markers were based on their availability, therapeutic activity and abundancy in a particular medicinal plant component. Moreover, an extensive literature search also helped in the selection of marker analytes, symbolic of a particular herb in the DSV formulation. The chief sweet-tasting triterpenoidal saponin of licorice, glycyrrhizin, represents 10% of the licorice root and glabridin, the chief isoflavone identified is found in the range of 0.08% and 0.35% [31]. Eugenol, the chief essential oil component (≈ 89%) is considered to be emblematic of clove [32]. Aerial parts of *Cressa cretica* are found to be rich in coumarins [10]. Phytochemical characterization of *Anacylus pyrethrum* showed the presence of cinnamic acid [33]. 6-Gingerol, the main bioactive component of ginger, was quantified and found to be 60.44 ± 2.53 mg/g of ginger extract [34]. Galls of *Pistacia integerrima* are reported to be rich in polyphenolics, i.e., gallic acid [35]. Chemical characterization of *Cinnamomum zeylamicum* bark revealed the presence of eugenol in appreciable amounts [36]. Besides, the herbal components, DSV formulation also contains seven different bhasma (Table 1). These are unique Ayurvedic herbo-mineral preparations, which are added to a formulation to provide optimal alkalinity, by neutralization of the harmful acids in the body. Moreover, these preparations are considered to be efficacious and non-toxic in nature [37]. Therapeutic efficacies of Tankan and Sphatika bhasma against diseases of the throat and palate are well documented [38]. Kapardak bhasma, Abhraka bhasma, Godanti bhasma and Mukta shukti bhasma are reported to have potential anti-inflammatory potential [14]. Praval pishti, processed coral calcium, is imbued with anti-inflammatory properties, moreover the preparation also confers benefits against coughs and related ailments [39]. Thus, the anti-inflammatory potentials of the herbo-mineral elements of the formulation might be beneficial to provide symptomatic relief in the current SARS-CoV-2 infectivity. It is worth mentioning that since the bhasma are inorganic compounds they exhibit poor solubility in the organic solvent methanol. Hence, these herbo-mineral preparations are not expected to hinder the current analytical strategy.

HPLC is a versatile, precise and foremost favored method among the accessible chromatographic strategies for herbal analysis [40]. HPLC frameworks hyphenated with a spectroscopic detector gives a readier data of the analytes present in a sample. [41]. Thus, chemical astuteness of DSV was investigated utilizing a validated HPLC procedure.

An analytical strategy can certainly be titled paramount which is capable of providing reliable, consistent and precise information, when performed by diverse investigators in different research environments. Optimization is much sought after for the accomplishment of consistent and repeatable outcomes. Validation plays an imperative part in fulfilling this objective [42]. Development of the HPLC method, constituted of several trial and error procedures for selection of a suitable mobile phase. Moreover, pharmacoepial and FDA requirements to achieve optimum resolution and specificity of the targeted analytes were also taken into consideration [43,44]. A few solvent compositions appeared to give longer run times, and some of them were incapable of resolving the targeted analytes at the same time. Finally, the best resolution of all the marker components was achieved using 0.1% orthophosphoric acid in water adjusted to pH 2.5 with diethylamine (solvent A) and 0.1% orthophosphoric acid in acetonitrile: water (88:12) adjusted to pH 2.5 (solvent B) with a gradient elution program. Chromatographic quality and analysis time is strongly dependent on the flow rate of the mobile phase passing through the column in unit time. The chromatograms, acquired with a flow rate of 1 mL/min appeared to give convincing partition of the analytes. Pronounced analytical outcomes are accomplished with the proper selection of wavelength. For that, it is exceptionally imperative to assess the absorption spectra of the compound intrigued. The wavelengths for the individual compounds were selected based on their λ_max_ as depicted in Figure 2. Notably, piperine shows an absorption maximum at 340 nm, but for the simplicity of the developed HPLC method we preferred to quantify the same at 278 nm. A good peak resolution relies on the choice of a suitable column. The best resolution of the targeted analytes was accomplished by employing a Shodex C18-4E (5 µm, 4.6 mm × 250 mm) column maintained at 35 °C and subjected to binary gradient elution.

Validation methods are established documented proofs that assure that the conditions selected for the strategy will reliably deliver consistent results. In addition, validation also considers the danger related with the components of a methodologically developed procedures by evaluating if the strategy is reproducible and scientifically sound [45]. These documented evidences further build certainty for the usage of the method. Thus, the developed HPLC method for the targeted analytes, gallic acid, protocatechuic acid, methyl gallate, ellagic acid, coumarin, cinnamic acid, glycyrrhizin, eugenol, 6-gingerol, piperine and glabridin were validated as per the ICH guidelines [20].

Specificity is the foremost essential parameter of any analytical procedure. It alludes to its capacity to produce a signal solely due to the analyte, in the presence of hindrances such as excipients, enantiomers and degradation products that are suspected to be present in the test framework [20]. The test should segregate the desired peak of analyte from other peaks of chromatogram. In this study, no peak was recognized near the retention times of the targeted analytes in standard solution when compared with a solvent blank. Thus, the developed HPLC method is specific for the determination of the targeted analytes in the tested DSV formulation. Limit of detection (LOD) and limit of quantification (LOQ) are the two vital performance characteristics in method validation. [20]. Signal to noise (S/N) is one of the classical methodologies for the determination of the above two important parameters. The concentration having signal to noise ratio 3:1 is referred as LOD and 10:1 as LOQ. The validation results revealed that the LOD and LOQ values for the targeted analytes were within the permissible limits, indicating the sensitivity of the developed analytical method. The linearity of an analytical method can be explained as its capability to show that the obtained test results are directly proportional to the analyte concentration within a given range. Correlation coefficient (*r*^2^) of 0.99 is an indicative of the linearity [20]. For HPLC, the calibration curves of all the targeted analytes exhibited good linear relationship *r*^2^ > 0.99. The residual analysis was performed on the individual targeted analytes (Appendix A). The smaller residual sum of square (RSS) values in comparison to the regression sum of squares further confirmed that the values obtained by plotting response vs concentration are linear [46]. Thus the proposed method is in the accordance with the ICH guidelines and appropriate for the simultaneous quantification of the desired signature compounds. The precision studies were conducted at two levels, repeatability (intraday precision), which signifies the precision under the same operating conditions over a short interval of time and intermediate precision (interday precision) which represents the precision on different days. [20]. The obtained RSD values of all the targeted analytes were found to be less than 2%, confirming that the developed method is sufficiently precise. The recovery refers to the percentage of the concentration of the targeted analyte in a sample [20]. The percentage recoveries of all the targeted analytes at the three different concentrations ranged from 92.75 to 100.13% demonstrating their good recovery. The results provided evidence that established HPLC–DAD method is accurate for simultaneous estimations of gallic acid, protocatechuic acid, methyl gallate, ellagic acid, coumarin, cinnamic acid, glycyrrhizin, eugenol, 6-gingerol, piperine and glabridin in DSV. The operational components in a research area tend to vary within a realistic range. Robustness studies aim to examine the influence of the potential sources of variations such as, flow rate and column temperature in the responses of the method. The robustness of an analytical strategy is the degree of its capacity to stay unaffected by small but deliberate variations in the method parameters, likely to happen amid the routine usage [20]. %RSD of all the eighteen determinations were found to be within the prescribed limits according to the ICH guidelines indicating the robustness of the method. Rugged strategies are the one that endures minor variation in test conditions, can be run effectively by any regular chromatographer, and does not essentially requires identical HPLC system for its use. Rugged methods are essentially trouble free and transferable [20]. The results indicated that %RSD of targeted analytes were within permissible range (NMT 10%) which indicated the ruggedness of the developed HPLC method.

The developed and validated HPLC–DAD method was further applied for the simultaneous estimation of gallic acid, protocatechuic acid, methyl gallate, ellagic acid, coumarin, cinnamic acid, glycyrrhizin, eugenol, 6-gingerol, piperine and glabridin in five different batches of DSV. Differences in the climatic as well as growing conditions of herbs often leads to the variability in the detected quantity of the secondary metabolites. For this quality assessment of the herbals utilizing a single marker is considered as a very vague approach. Thus, for qualitative check of botanicals, choice of multiple markers becomes rather vital. Hence, we confirm that the proposed analytical strategy is adequate, validated and pertinent for the quality control of DSV formulation.

## 4. Materials and Methods

### 4.1. Chemicals, Reagents and Samples

The AR grade solvents, toluene, ethyl acetate, formic acid, acetic acid and methanol (HPLC grade) were procured from Merck (Darmstad, Germany), acetonitrile from Honeywell (Dusseldorf, Germany) and deionized water was obtained from a Milli Q system (Millipore, Billerica, MA, USA). Authentic standards of gallic acid (Cat No. 91215, Sigma Aldrich, St. Louis, MO, USA), protocatechuic acid (Cat No. P006, Natural Remedies, Bangalore, Karnataka, India), coumarin (Cat No. C4261, Sigma Aldrich, St. Louis, MO, USA), cinnamic acid (Cat No. 29955, Sisco Research Lab, Mumbai, Maharashtra, India), eugenol (Cat No. 35995, Sigma Aldrich, St. Louis, MO, USA), 6-gingerol (Cat No. 11707, Cayman Chemical, Ann Arbor, MI, USA), piperine (Cat No. P49007-5G, Sigma Aldrich, St. Louis, MO, USA), glabridin (Cat No. G005, Natural Remedies, Bangalore, Karnataka, India), ellagic acid (Cat No. E2250, Cayman Chemical, Ann Arbor, MI, USA) and glycyrrhizin (Cat No. G008, Natural Remedies, Bangalore, Karnataka, India) were used for the analysis. Samples from five different batches of Divya-Swasari-Vati, (#B SWV117, #B SWV084, #A SWV023, #A SWV102 and #B SWV239) were used for the chemical analysis. DSV samples were sourced from Divya Pharmacy (Haridwar, India) and were stored in airtight bottles for further use.

### 4.2. Analytical Investigations

#### 4.2.1. UPLC/QToF MS Analysis

Preparation of DSV sample solution:

10 mL of methanol:water (80:20) was added to about 100 mg of powdered DSV sample and sonicated for 15 min. The sonicated solution was then centrifuged for 5 min at 5000 rpm and filtered using 0.22 μm nylon filter. The filtered DSV solution was further used for the analysis.

Instrumentation

Analysis was performed on a Xevo G2-XS QToF with Acquity UPLC-I Class and Unifi software (Waters Corporation, Milford, MA, USA). The main working parameters for mass spectrometry were set as follows, ionization type-ESI, mode-MS^E^, acquisition time-56 min, mass range (*m/z*)—50–1200 *m/z*, low collision energy—6 eV, high collision energy—20–40 eV (ramp), cone voltage—40 V, capillary voltage—1.5 kV (for positive mode), 2 kV (for negative mode), source temperature—120 °C, desolvation temperature—500 °C, cone gas flow—50 L/h, desolvation gas flow—900 L/h. Mass was corrected during acquisition, using an external reference (Lock–Spray) consisting of 0.2 ng/mL solution of leucine enkephalin infused at a flow rate of 10 µL/min via a lock–spray interface, generating a reference ion for the positive ion mode [(M + H)^+^
*m/z* 556.2766] and for the negative ion mode [(M − H)^−^
*m/z* 554.2620] to ensure mass correction during the MS analysis. The lock–spray scan time was set at 0.25 s with an interval of 30 s. The elution was carried out at a flow rate of 0.3 mL/min using gradient elution of mobile phase 0.1% formic acid in water (mobile phase A) and 0.1 % formic acid in acetonitrile (mobile phase B). The volume ratio of solvent B was changed as follows, 5–10% B for 0–5 min, 10–30% B for 5–15 min, 30–55% B for 15–25 min, 55–70% B for 25–40 min, 70–80% B for 40–50 min, 80–85% B for 50–55 min, 85–5% B for 55–56 min, 5% B for 56–60 min. A total of 2 µL of the test solution was injected for the screening and the chromatograph was recorded for 56 min.

Identification of marker components in DSV

Compounds were analyzed by their respective mass to charge ratio and fragmentation pattern. Mass/charge (*m/z*) ratio was selected based on the molecular ions of these compounds. Data acquisitions were collected under both positive (+ve) and negative (−ve) modes of ionization using full spectrum scan analysis. Further, the identified components were grouped in according to their optimum determination in each ionization mode.

#### 4.2.2. HPLC–DAD Method Development and Optimization

Preparation of standard solution:

Stock solutions of gallic acid, protocatechuic acid, methyl gallate, ellagic acid, coumarin, cinnamic acid, glycyrrhizin, eugenol, 6-gingerol, piperine and glabridin (1000 ppm) were prepared by dissolving accurately weighed standards in methanol. The stock solutions were mixed and diluted with methanol: water (80:20) to prepare the appropriate concentrations (50 ppm) for working standard solutions.

Preparation of DSV sample solution

Samples of 0.5 g of powdered DSV from batches #B SWV117, #B SWV084, #A SWV023, #A SWV102 and #B SWV239 were diluted with 10 mL water:methanol (20:80) and sonicated for 30 min. The sonicated solution was centrifuged for 5 min at 10,000 rpm and filtered through 0.45 µm nylon filter (Test solution) and used for the analysis of protocatechuic acid, coumarin, cinnamic acid, 6-gingerol and glabridin. Solution A was further diluted up to 10 times with the same solvent and used for the analysis of gallic acid, methyl gallate, eugenol, piperine, ellagic acid and glycyrrhizin.

Instrumentation and chromatographic conditions

Analysis was performed on HPLC equipment, Prominence-i LC-2030c 3D Plus (Shimadzu Corporation, Kyoto, Japan). Three different reversed phase columns, Shodex C18-4E (5 µm, 4.6 mm× 250 mm) column, Shim pack GIST-HP C18 (3 µm, 4 mm× 150 mm) column, Shim pack GIST-HP C18 (5 µm, 4.6 mm × 250 mm) column were evaluated during chromatographic optimization. Separation was achieved using a Shodex C18-4E (5 µm, 4.6 mm × 250 mm) column. Different mobile phase, including methanol–0.1% glacial acetic acid in water, acetonitrile–0.1% acetic acid in water, methanol–0.1% orthophosphoric acid in water, acetonitrile–0.1% phosphoric acid in water, and acetonitrile–0.2% formic acid, were tried, at different pH of the mobile phase were tried. Finally, the elution was carried out using binary gradient mode using the mobile phase composed of 0.1% orthophosphoric acid in water (pH 2.5) and diethylamine (solvent A) and 0.1% orthophosphoric acid in acetonitrile: water (88:12) (solvent B) in binary gradient mode. The volume ratio of solvent B was changed as follows, 5–10% B for 0–10 min, 10–35% B for 10–30 min, 35–50% B for 30–40 min, 50–75% B for 40–50 min, 75% B for 50–55 min, 75–85% B for 55–60 min, 85–5% B for 65–66 min, 5% B for 66–70 min. The effluent from the column was detected by a diode array detector and the detection wavelength was set at 278 nm for gallic acid, methyl gallate, protocatechuic acid, coumarin, cinnamic acid, eugenol, 6-gingerol, piperine and glabridin, whereas for ellagic acid and glycyrrhizin, the detection was carried out at 250 nm. The temperature of the column was kept at 35 °C and the sample injection volume was 10 µL. The method was optimized using a suitable solvent system and monitoring suitable wavelength for separation of components with the highest sensitivity. Other parameters with optimized injection volume, flow rate and column temperature were used for maximum resolution and short analysis time.

### 4.3. Method Validation

Eleven marker components, namely gallic acid, protocatechuic acid, methyl gallate, ellagic acid, coumarin, cinnamic acid, glycyrrhizin, eugenol, 6-gingerol, piperine and glabridin were validated using HPLC in DSV sample as per the recommendations of International Council on Harmonization (ICH) guidelines [20]. Specificity of an analytical procedure refers to its ability to unequivocally assess an analyte in the presence of the other components which may be expected to be present [20]. The specificity of the HPLC method was evaluated to ensure that there was no interference between the solvent blank and standard solution. The specificity was studied by injecting 10 μL solutions of blank at 278 nm and 250 nm respectively. The linearity of an analytical procedure is an important parameter which signifies to its ability to produce the test results that are directly proportional to the concentration of an analyte in a given concentration range [20]. To evaluate the linearity and range of the developed method eleven different standard solutions for each of the targeted analytes were prepared in different concentration ranges (0.15–100 µg/g) by diluting the stock solutions with methanol. The calibration curves were constructed by plotting the peak area of standards versus respective concentrations. The degree of linearity was estimated by calculating correlation coefficient, using the calibration curve. The limit of detection (LOD) is described as the lowest concentration of the analyte in a sample which can be reliably detected but not necessarily quantitated by a particular analytical method. Whereas, the limit of quantification (LOQ) is considered as the lowest concentration of the analyte which can be quantitatively determined with suitable precision and accuracy [20]. LOD and LOQ of each marker component were determined based on signal-to-noise method (S/N ratio). S/N ratio for LOD was performed by injecting 6 replicates of minimum concentration at which the component was reliably detected, similarly LOQ was performed by injecting six replicates of a concentration at which the analyte can be reliably quantified. Moreover, the limit of peak area %RSD for LOD and LOQ was set at NMT 33% and NMT 10% respectively. The parameter precision expresses the degree of scatter between a series of measurements obtained from a multiple sampling of the homogeneous sample [20]. The intraday (repeatability) and interday (intermediate precision) precision (*n* = 6) was evaluated by calculating the relative standard deviation (%RSD) with accuracy in the quantification of the sample set. Accuracy of an analytical procedure refers to the closeness of the agreement between the value which is true and the experimental value [20]. The accuracy of the developed method was thoroughly evaluated by recovery studies. Analytical recovery was performed by spiking DSV sample with the reference standards at known concentration levels, such as 80%, 100% and 120% as per the area ratio method. Recoveries at three different concentrations were thus calculated. Robustness of the method provides an indication of its reliability during normal usage [20]. Robustness of method performance was verified by incorporating small intentional changes in the experimental parameters for example column temperature, and flow rate. Obtained data for each case was evaluated by calculating %RSD. Ruggedness of the current method was confirmed by testing the reproducibility of the test results under the variation in operational conditions by different analysts on different days to assure for any changes in the result. The percentage RSD for the retention area was calculated.

### 4.4. Quantitative Analysis of Targeted Analytes in Five Different Batches of DSV

For assuring the reliability of the developed and validated method quantitative analysis of gallic acid, protocatechuic acid, methyl gallate, ellagic acid, coumarin, cinnamic acid, glycyrrhizin, eugenol, 6-gingerol, piperine and glabridin was carried out in different batches of DSV. Quantitative analysis of particular targeted analyte was carried out against its reference standard by calculating area under the peak of analyte, in HPLC chromatogram.

### 4.5. Data Analysis

Statistical analyses were performed using Graph Pad Prism 7.0 (GraphPad Software, Inc., San Diego, CA, USA). Characterization of the marker analytes was performed using the Unifi software (Waters Corporation).

## 5. Conclusions

The analysis and quality control of traditional herbal medicines is heading in the direction of extensive and comprehensive research for uncovering their inalienable complexities. The present inventive research is an attempt to outline the applicability of two state-of-art chromatographic techniques, UPLC/QToF MS and HPLC–DAD on the quality of the calcio-herbal formulation Divya-Swasari-Vati (DSV). Seventy four phytometabolites were identified in the formulation using UPLC/QToF MS. Further, the simultaneous analysis of the selected markers—gallic acid, protocatechuic acid, methyl gallate, ellagic acid, coumarin, cinnamic acid, glycyrrhizin, eugenol, 6-gingerol, piperine and glabridin—in five different batches of DSV was carried out using the novel validated HPLC method. The established method was rapid, simple and reliable for simultaneous quantitative estimation of eleven marker components in Divya-Swasari-Vati. These outcomes may also assist in analysis of other extracts and formulations, having similar marker profiles.

## Figures and Tables

**Figure 1 pharmaceuticals-14-00297-f001:**
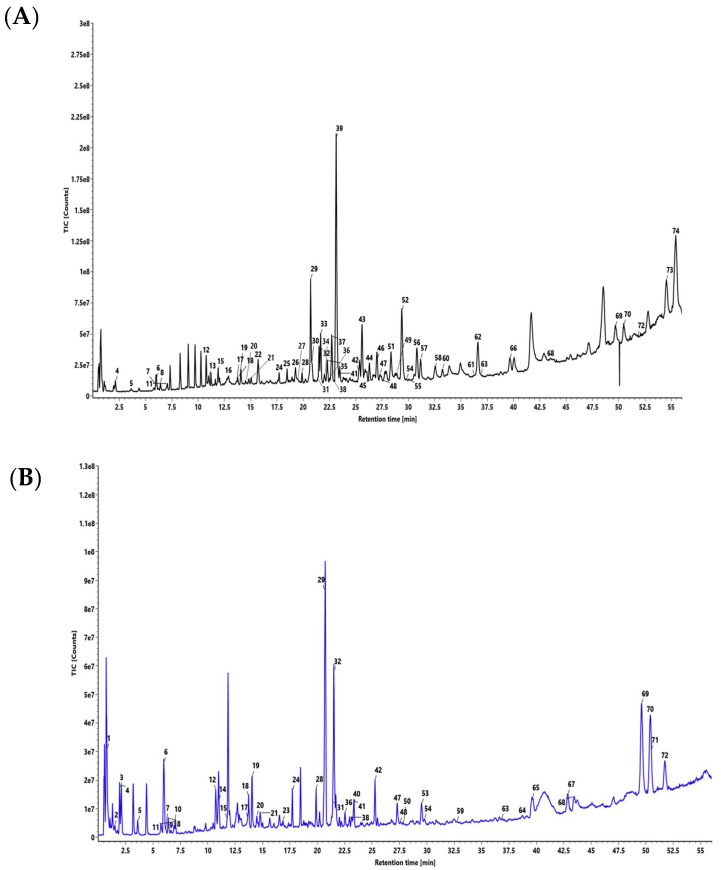
Total ion chromatogram of seventy-four compounds characterized in Divya-Swasari-Vati (DSV) in (**A**) positive mode and (**B**) negative mode using UPLC/QToF MS. The seventy-four compounds are, (1) quinic acid, (2) galloyl glucose, (3) gallic acid, (4) Theogallin, (5) protocaechuic acid, (6) methyl gallate, (7) 3, 4-di-O-galloylquinic acid, (8) chlorogenic acid, (9) 1,6-di-O-galloyl-glucose, (10) digallic acid, (11) cryptochlorogenic acid, (12) neoliquiritrin, (13) liquiritigenin, (14) ellagic acid, (15) quercetin-3-O-β-d-glucuronide, (16) coumarin, (17) kushenol O, (18) licurazide, (19) liquiritin apioside, (20) liquiritrin, (21) N-feruloyltyramine, (22) cinnamic acid, (23) 24-hydroxy licoricesaponin A3, (24) licoricesaponin A3 (25) glabrolide, (26) eugenol, (27) piperanine, (28) licoricesaponin G2, (29) glycyrrhizin, (30) piperyline, (31) 3-o-(β-d-glucoronopyranosyl (1-2)-β-d-galacto pyranosyl) glycyrrhetic acid, (32) licoricesaponin K2, (33) 6-gingerol, (34) 4,5-dihydropiperlonguminine, (35) piperlonguminine, (36) licoricesaponin J2, (37) feruperine, (38) licoricesaponin C2, (39) piperine, (40) shinpterocarpin, (41) licoricesaponin B2, (42) glabridin, (43) piperettine, (44) piperolein A, (45) dipiperamide E, (46) retrofractamide A, (47) glabrol, (48) 1- methoxyphaseollidin, (49) piperolactam-C9:1 (8E), (50) 1-methoxyphaseollin, (51) dehydropipernonaline, (52) pipernonaline, (53) 2-αhydroxyursolic acid, (54) licochalcone A, (55) dipiperamide-D, (56) piperolein B, (57) pipercide, (58) 10,11-dihydropipercide, (59) sophoranodichromane D, (60) piperundecalidine, (61) shinflavanone, (62) guineesine, (63) glycyrrhetic acid, (64) ursolic acid, (65) glycyrrhetol, (66) liquidambronal, (67) betulonic acid, (68) oleanonic acid, (69) deoxyglabrolide, (70) glypallidifloric acid, (71) 5-hydroxyeicosatetraenoic acid, (72) ginkgolic acid, (73) N-isobutyl-(2E,4E)-octadecadienamide, (74) pipnoohine.

**Figure 2 pharmaceuticals-14-00297-f002:**
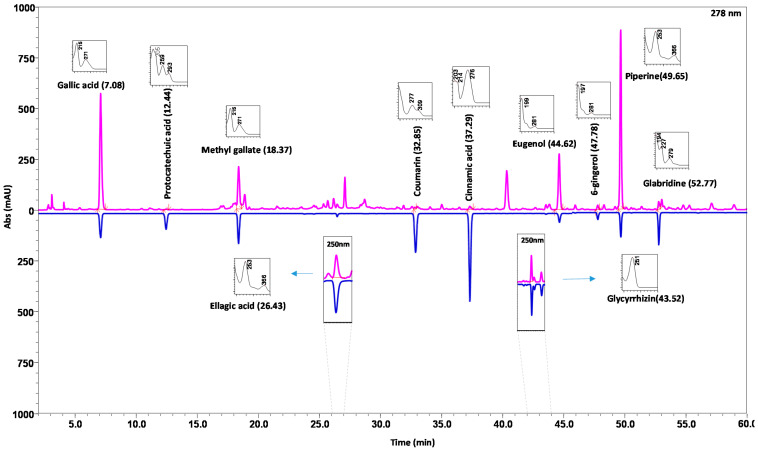
HPLC–DAD analysis identified and quantified the presence of 11 targeted marker components in DSV. The DSV sample (pink lines) was compared using reference standard mix (blue lines). The chromatograms were recorded at 278 nm for (methyl gallate, coumarin, cinnamic acid, eugenol, 6-gingerol, piperine and glabridin), and at 250 nm for ellagic acid and glycyrrhizin. UV-spectra of each detected analyte has been shown in the respective insets, along with HPLC retention times.

**Figure 3 pharmaceuticals-14-00297-f003:**
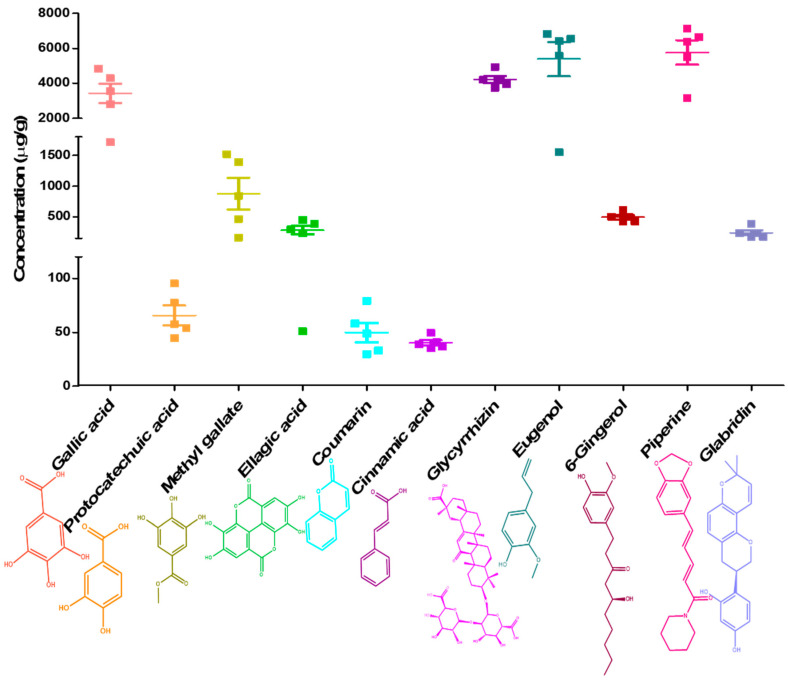
Quantitative analysis of gallic acid, protocatechuic acid, methyl gallate, ellagic acid, coumarin, cinnamic acid, glycyrrhzin, eugenol, 6-gingerol, piperine and glabridin using HPLC-DAD analysis in five different batches of Divya-Swasari-Vati (DSV). Scatter plot show detected concentrations of each analyte with mean and SEM (*n* = 5), in DSV formulation. Chemical structure of analytes have been sourced from www.pubchem.com (accessed on 22 March 2021).

**Table 1 pharmaceuticals-14-00297-t001:** Ingredients and Composition of Divya-Swasari-Vati (DSV) tablet formulation. Excipients: gum acacia (*Acacia arabica*) 4.62%, hydrated magnesium silicate 1.38% and colloidal silicon dioxide 1.38% are also present in the formulation.

S. No.	DSV Constituent’s Scientific Name	Hindi Vernacular Name	% in Each DSV Tablet
1	*Pistacia integerrima*	Kakadasingi	11.66
2	*Glycyrrhiza glabra*	Mulethi	11.85
3	*Cressa cretica*	Rudanti	11.66
4	*Piper nigrum*	Marich	7.77
5	*Piper longum*	Choti pippal	7.77
6	*Zingiber officinale*	Sounth	7.77
7	*Cinnamomum zeylanicum*	Dalchini	5.92
8	*Syzygium aromaticum*	Lavang	5.92
9	*Anacylus pyrethrum*	Akarkara	5.92
10	Herbally processed ash from calcined shell of pearl oyster (*Pinctada fucata*)	Mukta- Shukti Bhasma	2.33
11	Herbally processed ash from rich gypsum	Godanti Bhasma	2.33
12	Herbally processed ash from calcined cowry shell of *Cypraea moneta*	Kapardak Bhasma	2.33
13	Herbally processed ash from calcined mica	Abharak Bhasma	2.33
14	Herbally processed ash from calcined form of alum	Sphatika Bhasma	2.33
15	Coral calcium powder processed with rose water	Praval Pishti	2.33
16	Herbally processed ash from calcined borax	Tankan Bhasma	2.33

**Table 2 pharmaceuticals-14-00297-t002:** Identified metabolites in Divya-Swasari-Vati (DSV) on UPLC/QToF MS analysis.

Peak	Analyte	Formula	Neutral Mass (D)	Observed Mass (D)	RT (min)	Mode	Fragments
1	Quinic acid	C_7_H_12_O_6_	192.0634	191.0555	0.80	−ve	[C_7_H_12_O_6_]^−H^, *m/z* 173.0445, *m/z* 149.0443, *m/z* 129.0184, *m/z* 113.0258, *m/z* 89.0267
2	Galloylglucose	C_13_H_16_O_10_	332.0744	331.0665	1.50	−ve	[C_13_H_16_O_10_]^−H^, *m/z* 271.0442, *m/z* 211.0231, *m/z* 169.0130, *m/z* 151.0026
3	Gallic acid	C_7_H_6_O_5_	170.0215	169.0136	1.95	−ve	[C_7_H_6_O_5_]^−H^, *m/z* 153.0177, *m/z* 137.0238, *m/z* 125.0238
4	Theogallin	C_14_H_16_O_10_	344.0744	345.0821	2.13	+ve	[C_14_H_16_O_10_]^+H^, *m/z* 327.0714, *m/z* 247.0211, *m/z* 192.0607, *m/z* 153.0187, *m/z* 125.0239
				343.0667	2.00	−ve	[C_14_H_16_O_10_]^−H^, *m/z* 297.0600, *m/z* 271.0448, *m/z* 191.0550, *m/z* 166.9973, *m/z* 123.0092
5	Protocatechuic acid	C_7_H_6_O_4_	154.0266	155.0340	3.65	+ve	[C_7_H_6_O_4_]^+H^, *m/z* 137.0237
				153.0185	3.61	−ve	[C_7_H_6_O_4_]^−H^
6	Methyl gallate	C_8_H_8_O_5_	184.0372	185.0447	6.04	+ve	[C_8_H_8_O_5_]^+H^, *m/z* 169.0107, *m/z* 153.0186, *m/z* 139.0408
				183.0292	5.99	−ve	[C_8_H_8_O_5_]^−H^, *m/z* 168.0051, *m/z* 153.0181, *m/z* 124.0160, *m/z* 123.0079, *m/z* 106.0077
7	3,4-Di-O-galloylquinic acid	C_21_H_20_O_14_	496.0853	497.0923	6.24	+ve	[C_21_H_20_O_14_]^+H^, *m/z* 327.0702, *m/z* 247.0232, *m/z* 153.0186, *m/z* 139.0408
				495.0775	6.18	−ve	[C_21_H_20_O_14_]^−H^, *m/z* 343.0652, *m/z* 245.0076, *m/z* 191.0547, *m/z* 166.9966
8	Chlorogenic acid	C_16_H_18_O_9_	354.0951	355.1026	6.43	+ve	[C_16_H_18_O_9_]^+H^, *m/z* 319.0814, *m/z* 235.0602, *m/z* 205.0496, *m/z* 163.0395, *m/z* 130.0664
				353.0874	6.39	−ve	[C_16_H_18_O_9_]^−H^, *m/z* 275.0537, *m/z* 233.0444, *m/z* 205.0495, *m/z* 163.0388
9	1,6-Di-O-galloyl-glucose	C_20_H_20_O_14_	484.0853	483.0775	6.64	−ve	[C_20_H_20_O_14_]^−H^, *m/z* 313.0547, *m/z* 271.0442, *m/z* 169.0129, *m/z* 169.0050
10	Digallic acid	C_14_H_10_O_9_	322.0325	321.0246	6.94	−ve	[C_14_H_10_O_9_]^−H^, *m/z* 275.0173, *m/z* 257.0064, *m/z* 169.0130, *m/z* 168.0047, *m/z* 125.0237
11	Cryptochlorogenic acid	C_16_H_18_O_9_	354.0951	355.1028	7.08	+ve	[C_16_H_18_O_9_]^+H^, *m/z* 319.0818, *m/z* 301.0712, *m/z* 235.0606, *m/z* 217.0499, *m/z* 149.0238
				353.0873	7.05	−ve	[C_16_H_18_O_9_]^−H^, *m/z* 335.0735, *m/z* 233.0442, *m/z* 217.0489, *m/z* 217.0489, *m/z* 191.0324, *m/z* 147.0429
12	Neoliquiritin	C_21_H_22_O_9_	418.1264	419.1343	10.76	+ve	[C_21_H_22_O_9_]^+H^, *m/z* 389.1238, *m/z* 285.0760, *m/z* 257.0813, *m/z* 191.0330, *m/z* 137.0238, *m/z* 133.0863
				417.1192	10.73	−ve	[C_21_H_22_O_9_]^−H^, *m/z* 399.1010, *m/z* 297.0736, *m/z* 255.0651, *m/z* 254.0565, *m/z* 191.0328, *m/z* 135.0079
13	Liquiritigenin	C_15_H_12_O_4_	256.0736	257.0814	11.03	+ve	[C_15_H_12_O_4_]^+H^, *m/z* 239.0707, *m/z* 215.0715, *m/z* 163.0399, *m/z* 137.0239, *m/z* 119.0498
14	Ellagic acid	C_14_H_6_O_8_	302.0063	300.9986	11.03	−ve	[C_14_H_6_O_8_]^−H^, *m/z* 283.9943, *m/z* 178.9969, *m/z* 151.0027, *m/z* 135.0080
15	Quercetin-3-O-β-d-glucuronide	C_21_H_18_O_13_	478.0747	479.0826	11.81	+ve	[C_21_H_18_O_13_]^+H^, *m/z* 303.0506, *m/z* 245.0452, *m/z* 147.0448
				477.0677	11.77	−ve	[C_21_H_18_O_13_]^−H^, *m/z* 301.0336, *m/z* 299.0180, *m/z* 243.0281, *m/z* 151.0025
16	Coumarin	C_9_H_6_O_2_	146.0368	147.0446	12.88	+ve	[C_9_H_6_O_2_]^+H^, *m/z* 131.0499
17	Kushenol O	C_27_H_30_O_13_	562.1686	563.1763	13.67	+ve	[C_27_H_30_O_13_]^+H^, *m/z* 549.1600, *m/z* 387.1322, *m/z* 269.0813, *m/z* 237.0543, *m/z* 153.0719
				561.1619	13.65	−ve	[C_27_H_30_O_13_]^−H^, *m/z* 547.1428, *m/z* 401.0868, *m/z* 267.0648, *m/z* 252.0410, *m/z* 151.0391
18	Licurazide	C_26_H_30_O_13_	550.1686	551.1762	13.77	+ve	[C_26_H_30_O_13_]^+H^, *m/z* 461.1421, *m/z* 419.1335, *m/z* 317.0667, *m/z* 257.0812, *m/z* 239.0705, *m/z* 137.0238
				549.1616	13.74	−ve	[C_26_H_30_O_13_]^−H^, *m/z* 417.1159, *m/z* 357.0962, *m/z* 255.0650, *m/z* 254.0566, *m/z* 135.0082
19	Liquiritin apioside	C_26_H_30_O_13_	550.1686	551.1757	14.07	+ve	[C_26_H_30_O_13_]^+H^, *m/z* 453.1153, *m/z* 419.1333, *m/z* 389.1236, *m/z* 269.0813, *m/z* 257.0813, *m/z* 137.0238
				549.1614	14.04	−ve	[C_26_H_30_O_13_]^−H^, *m/z* 533.1630, *m/z* 399.1061, *m/z* 255.0651, *m/z* 165.0549, *m/z* 135.008
20	Liquiritin	C_21_H_22_O_9_	418.1264	419.1344	14.51	+ve	[C_21_H_22_O_9_]^+H^, *m/z* 355.1184, *m/z* 257.0811, *m/z* 255.0655, *m/z* 147.0446
				417.1191	14.47	−ve	[C_21_H_22_O_9_]^−H^, *m/z* 343.1189, *m/z* 299.0544, *m/z* 255.0650, *m/z* 253.0490, *m/z* 163.0387, *m/z* 135.0079
21	N-feruloyltyramine	C_18_H_19_NO_4_	313.1314	314.1395	14.83	+ve	[C_18_H_19_NO_4_]^+H^, *m/z* 177.0552, *m/z* 145.0289, *m/z* 121.0652
				312.1240	14.80	−ve	[C_18_H_19_NO_4_]^−H^, *m/z* 297.0988, *m/z* 178.0501, *m/z* 148.0520
22	Cinnamic acid	C_9_H_8_O_2_	148.0524	149.0603	15.71	+ve	[C_9_H_8_O_2_]^+H^, *m/z* 131.0498
23	24-Hydroxy-licoricesaponin A3	C_48_H_72_O_22_	1000.4515	999.4485	16.86	−ve	[C_48_H_72_O_22_]^−H^, *m/z* 939.4566, *m/z* 819.3776, *m/z* 485.3237, *m/z* 373.1632, *m/z* 179.0701
24	Licoricesaponin A3	C_48_H_72_O_21_	984.4566	985.4633	17.71	+ve	[C_48_H_72_O_21_]^+H^, *m/z* 866.3528, *m/z* 809.4295, *m/z* 615.3875, *m/z* 453.3357, *m/z* 435.3246, *m/z* 153.0184
				983.4525	17.72	−ve	[C_48_H_72_O_21_]^−H^, *m/z* 645.3610, *m/z* 469.3300, *m/z* 351.0545, *m/z* 193.0348
25	Glabrolide	C_30_H_44_O_4_	468.3240	469.3319	18.46	+ve	[C_30_H_44_O_4_]^+H^, *m/z* 439.3570, *m/z* 405.3154, *m/z* 315.1961, *m/z* 233.1539, *m/z* 175.1485, *m/z* 149.1327
26	Eugenol	C_10_H_12_O_2_	164.0837	164.0838	19.26	+ve	[C_10_H_12_O_2_]^-e^, *m/z* 149.0603, *m/z* 131.0498, *m/z* 119.0497
27	Piperanine	C_17_H_21_NO_3_	287.1521	288.1608	19.40	+ve	[C_17_H_21_NO_3_]^+H^, *m/z* 256.1340, *m/z* 203.0709, *m/z* 171.0440, *m/z* 137.0604
28	Licoricesaponin G2	C_42_H_62_O_17_	838.3987	839.4069	19.88	+ve	[C_42_H_62_O_17_]^+H^, *m/z* 582.2634, *m/z* 487.3414, *m/z* 469.3309, *m/z* 189.1641, *m/z* 175.1484
				837.3944	19.89	−ve	[C_42_H_62_O_17_]^−H^, *m/z* 793.3981, *m/z* 623.2339, *m/z* 431.2272, *m/z* 351.0551, *m/z* 193.0342
29	Glycyrrhizin	C_42_H_62_O_16_	822.4038	823.4115	20.71	+ve	[C_42_H_62_O_16_]^+H^, *m/z* 700.4142, *m/z* 647.3781, *m/z* 453.3364, *m/z* 435.3262, *m/z* 272.1290, *m/z*, 189.1645
				821.3994	20.69	−ve	[C_42_H_62_O_16_]^−H^, *m/z* 759.3939, *m/z* 645.3619, *m/z* 499.3038, *m/z* 351.0555, *m/z* 193.0348
30	Piperyline	C_16_H_17_NO_3_	271.1208	272.1293	20.84	+ve	[C_16_H_17_NO_3_]^+H^, *m/z* 244.1349, *m/z* 242.1165, *m/z* 201.0551, *m/z* 171.0447, *m/z* 135.0449, *m/z* 122.0360
31	3-O-(β-d-Glucuronopyranosyl-(1-2)-β-d-galactopyranosyl)glycyrrhetic acid	C_42_H_64_O_15_	808.4245	809.4319	21.41	+ve	[C_42_H_64_O_15_]^+H^, *m/z* 633.3987, *m/z* 439.3571, *m/z* 437.3407, *m/z* 241.0879, *m/z* 175.1114
				807.4197	21.42	−ve	[C_42_H_64_O_15_]^−H^, *m/z* 745.4132, *m/z* 485.3251, *m/z* 303.2322, *m/z* 187.0961
32	Licoricesaponine K2	C_42_H_62_O_16_	822.4038	823.4114	21.51	+ve	[C_42_H_62_O_16_]^+H^, *m/z* 700.4185, *m/z* 647.3779, *m/z* 453.3364, *m/z* 435.3259, *m/z* 235.1698, *m/z* 189.1644
				821.3991	21.52	−ve	[C_42_H_62_O_16_]^−H^, *m/z* 807.4142, *m/z* 645.3607, *m/z* 485.3251, *m/z* 351.0550, *m/z* 193.0344
33	6-Gingerol	C_17_H_26_O_4_	294.1831	317.1738	21.66	+ve	[C_17_H_26_O_4_]^+Na^, *m/z* 259.1702, *m/z* 177.0917, *m/z* 162.0680, *m/z* 137.0605
34	4,5-Dihydropiperlonguminine	C_16_H_21_NO_3_	275.1521	276.1604	22.03	+ve	[C_16_H_21_NO_3_]^+H^, *m/z* 246.1507, *m/z* 203.0712, *m/z* 135.0446, *m/z* 131.0494
35	Piperlonguminine	C_16_H_19_NO_3_	273.1365	274.1448	22.29	+ve	[C_16_H_19_NO_3_]^+H^, *m/z* 262.1438, *m/z* 201.0549, *m/z* 171.0446, *m/z* 135.0447, *m/z* 115.0992
36	Licoricesaponine J2	C_42_H_64_O_16_	824.4194	825.4265	22.53	+ve	[C_42_H_64_O_16_]^+H^, *m/z* 613.3720, *m/z* 455.3516, *m/z* 409.3463, *m/z* 205.1061
				823.4147	22.53	−ve	[C_42_H_64_O_16_]^−H^, *m/z* 761.4095, *m/z* 597.2575, *m/z* 439.1797, *m/z* 351.0551, *m/z* 193.0346, *m/z* 175.0214
37	Feruperine	C_17_H_21_NO_3_	287.1521	288.1602	22.72	+ve	[C_17_H_21_NO_3_]^+H^, *m/z* 270.1496, *m/z* 217.1090, *m/z* 203.0709, *m/z* 135.0447, *m/z* 124.0768
38	Licoricesaponin C2	C_42_H_62_O_15_	806.4089	829.3991	22.94	+ve	[C_42_H_62_O_15_]^+Na^, *m/z* 560.3732, *m/z* 437.3411, *m/z* 396.2542, *m/z* 285.1852, *m/z* 173.0946
				805.4042	22.95	−ve	[C_42_H_62_O_15_]^−H^, *m/z* 743.3975, *m/z* 645.3662, *m/z* 501.3191, *m/z* 351.0552, *m/z* 167.0342
39	Piperine	C_17_H_19_NO_3_	285.1365	286.1449	23.13	+ve	[C_17_H_19_NO_3_]^+H^, *m/z* 258.1495, *m/z* 201.0552, *m/z* 171.0447, *m/z* 135.0449, *m/z* 112.0763
40	Shinpterocarpin	C_20_H_18_O_4_	322.1205	321.1135	23.28	−ve	[C_20_H_18_O_4_]^−H^, *m/z* 306.0883, *m/z* 265.0490, *m/z* 237.0542, *m/z* 175.0758, *m/z* 145.0290
41	Licoricesaponin B2	C_42_H_64_O_15_	808.4245	831.4131	23.34	+ve	[C_42_H_64_O_15_]^+Na^, *m/z* 731.3659, *m/z* 602.2705, *m/z* 485.3259, *m/z* 439.3567, *m/z* 279.1421, *m/z* 213.1123
				807.4201	23.35	−ve	[C_42_H_64_O_15_]^−H^, *m/z* 779.4222, *m/z* 695.3628, *m/z* 473.2729, *m/z* 351.0551, *m/z* 193.0343
42	Glabridin	C_20_H_20_O_4_	324.1362	325.1445	25.28	+ve	[C_20_H_20_O_4_]^+H^, *m/z* 309.1130, *m/z* 270.0883, *m/z* 189.0916, *m/z* 173.0606, *m/z* 123.0447
				323.1292	25.26	−ve	[C_20_H_20_O_4_]^−H^, *m/z* 308.1037, *m/z* 268.0723, *m/z* 201.0915, *m/z* 135.0449
43	Piperettine	C_19_H_21_NO_3_	311.1521	312.1605	25.59	+ve	[C_19_H_21_NO_3_]^+H^, *m/z* 294.1501, *m/z* 227.0709, *m/z* 197.0603, *m/z* 161.0602, *m/z* 138.0920
44	Piperolein A	C_19_H_25_NO_3_	315.1834	316.1921	26.29	+ve	[C_19_H_25_NO_3_]^+H^, *m/z* 231.1025, *m/z* 194.1547, *m/z* 135.0448, *m/z* 131.0497
45	Dipiperamide E	C_34_H_38_N_2_O_6_	570.2730	571.2809	26.41	+ve	[C_34_H_38_N_2_O_6_]^+H^, *m/z* 444.1771, *m/z* 286.1444, *m/z* 201.0520, *m/z* 173.0559
46	Retrofractamide A	C_20_H_25_NO_3_	327.1834	328.1919	27.05	+ve	[C_20_H_25_NO_3_]^+H^, *m/z* 227.1072, *m/z* 187.0758, *m/z* 161.0602, *m/z* 131.0498
47	Glabrol	C_25_H_28_O_4_	392.1988	393.2070	27.31	+ve	[C_25_H_28_O_4_]^+H^, *m/z* 337.1442, *m/z* 321.1129, *m/z* 281.0814, *m/z* 203.0708, *m/z* 149.0240, *m/z* 137.0604
				391.1922	27.29	−ve	[C_25_H_28_O_4_]^−H^, *m/z* 203.0707, *m/z* 187.1122, *m/z* 132.0577
48	1-Methoxyphaseollidin	C_21_H_22_O_5_	354.1467	355.1551	27.58	+ve	[C_21_H_22_O_5_]^+H^, *m/z* 265.0494, *m/z* 189.0912, *m/z* 153.0557
				353.1397	27.55	−ve	[C_21_H_22_O_5_]^−H^, *m/z* 295.0591, *m/z* 201.0911, *m/z* 150.0315
49	Piperolactam-C9:1(8E)	C_20_H_27_NO_3_	329.1991	330.2071	27.81	+ve	[C_20_H_27_NO_3_]^+H^, *m/z* 259.1323, *m/z* 208.1702, *m/z* 135.0446, *m/z* 133.0650
50	1-Methoxyphaseollin	C_21_H_20_O_5_	352.1311	351.1239	27.86	−ve	[C_21_H_20_O_5_]^−H^, *m/z* 321.1108, *m/z* 267.0644, *m/z* 201.0913, *m/z* 146.0356
51	Dehydropipernonaline	C_21_H_25_NO_3_	339.1834	340.1915	28.34	+ve	[C_21_H_25_NO_3_]^+H^, *m/z* 286.1445, *m/z* 227.1071, *m/z* 179.1310, *m/z* 161.0602, *m/z* 112.0761
52	Pipernonaline	C_21_H_27_NO_3_	341.1991	342.2072	29.38	+ve	[C_21_H_27_NO_3_]^+H^, *m/z* 314.2119, *m/z* 229.1227, *m/z* 161.0601, *m/z* 135.0447, *m/z* 112.0761
53	2α-Hydroxyursolic acid	C_30_H_48_O_4_	472.3553	471.3488	29.52	−ve	[C_30_H_48_O_4_]^−H^, *m/z* 423.3237, *m/z* 393.3123, *m/z* 279.2320
54	Licochalcone A	C_21_H_22_O_4_	338.1518	339.1600	29.82	+ve	[C_21_H_22_O_4_]^+H^, *m/z* 276.0771, *m/z* 229.1227, *m/z* 189.0913, *m/z* 137.0602
				337.1449	29.79	−ve	[C_21_H_22_O_4_]^−H^, *m/z* 322.1187, *m/z* 267.0662, *m/z* 201.0910, *m/z* 175.0756, *m/z* 134.0369
55	Dipiperamide D	C_36_H_40_N_2_O_6_	596.2886	597.2961	30.18	+ve	[C_36_H_40_N_2_O_6_]^+H^, *m/z* 512.2070, *m/z* 334.1427, *m/z* 286.1444, *m/z* 186.0655
56	Piperolein B	C_21_H_29_NO_3_	343.2147	344.2230	30.81	+ve	[C_21_H_29_NO_3_]^+H^, *m/z* 286.1447, *m/z* 222.1860, *m/z* 154.1234, *m/z* 135.0448
57	Pipercide	C_22_H_29_NO_3_	355.2147	356.2231	31.16	+ve	[C_22_H_29_NO_3_]^+H^, *m/z* 283.1334, *m/z* 255.1387, *m/z* 234.1858, *m/z* 135.0448, *m/z* 133.1014
58	10,11-Dihydropipercide	C_22_H_31_NO_3_	357.2304	358.2385	32.50	+ve	[C_22_H_31_NO_3_]^+H^, *m/z* 285.1489, *m/z* 191.1066, *m/z* 135.0445
59	Sophoranodichromane D	C_25_H_28_O_5_	408.1937	407.1865	32.73	−ve	[C_25_H_28_O_5_]^−H^, *m/z* 350.1141, *m/z* 203.1064, *m/z* 203.0696, *m/z* 148.0522
60	Piperundecalidine	C_23_H_29_NO_3_	367.2147	368.2232	33.25	+ve	[C_23_H_29_NO_3_]^+H^, *m/z* 340.2281, *m/z* 255.1386, *m/z* 215.1071, *m/z* 135.0447, *m/z* 133.1011
61	Shinflavanone	C_25_H_26_O_4_	390.1831	391.1912	36.31	+ve	[C_25_H_26_O_4_]^+H^, *m/z* 375.1594, *m/z* 257.0773, *m/z* 215.1072, *m/z* 189.0914, *m/z* 147.0810
62	Guineesine	C_24_H_33_NO_3_	383.2460	384.2543	36.61	+ve	[C_24_H_33_NO_3_]^+H^, *m/z* 311.1648, *m/z* 283.1702, *m/z* 257.1535, *m/z* 175.0757, *m/z* 135.0447, *m/z* 131.0497
63	Glycyrrhetic acid	C_30_H_46_O_4_	470.3396	471.3471	36.90	+ve	[C_30_H_46_O_4_]^+H^, *m/z* 407.3320, *m/z* 364.3158, *m/z* 229.1937, *m/z* 175.1489, *m/z* 173.1333
				469.3325	36.85	−ve	[C_30_H_46_O_4_]^−H^, *m/z* 451.3185, *m/z* 407.3289
64	Ursolic acid	C_30_H_48_O_3_	456.3604	455.3538	38.72	−ve	[C_30_H_48_O_3_]^−H^, *m/z* 389.3044, *m/z* 331.2605, *m/z* 125.0969
65	Glycyrrhetol	C_30_H_48_O_3_	456.3604	455.3538	39.61	−ve	[C_30_H_48_O_3_]^−H^, *m/z* 407.3301
66	Liquidambronal	C_30_H_46_O_2_	438.3498	439.3578	39.68	+ve	[C_30_H_46_O_2_]^+H^, *m/z* 408.3381, *m/z* 297.2555, *m/z* 255.2120, *m/z* 203.1800, *m/z* 191.1800, *m/z* 135.1173
67	Betulonic acid	C_30_H_46_O_3_	454.3447	453.3387	42.87	−ve	[C_30_H_46_O_3_]^−H^, *m/z* 301.2136, *m/z* 247.2058
68	Oleanonic acid	C_30_H_46_O_3_	454.3447	455.3511	43.51	+ve	[C_30_H_46_O_3_]^+H^, *m/z* 409.3453, *m/z* 343.2649, *m/z* 261.2222, *m/z* 203.1799, *m/z* 177.1643
				453.3384	43.44	−ve	[C_30_H_46_O_3_]^−H^, *m/z* 422.2805
69	Deoxyglabrolide	C_30_H_46_O_3_	454.3447	455.3522	49.70	+ve	[C_30_H_46_O_3_]^+H^, *m/z* 437.3415, *m/z* 353.2489, *m/z* 321.2565, *m/z* 215.1799, *m/z* 189.1644, *m/z* 161.1330
				453.3387	49.60	−ve	[C_30_H_46_O_3_]^−H^, *m/z* 393.3134, *m/z* 317.2845, *m/z* 245.1536, *m/z* 177.0910, *m/z* 153.1281
70	Glypallidifloric acid	C_30_H_46_O_3_	454.3447	455.3521	50.49	+ve	[C_30_H_46_O_3_]^+H^, *m/z* 437.3417, *m/z* 353.2487, *m/z* 297.2582, *m/z* 203.1800, *m/z* 161.1330, *m/z* 135.1175
				453.3388	50.40	−ve	[C_30_H_46_O_3_]^−H^, *m/z* 393.3133, *m/z* 167.1100
71	5-Hydroxyeicosatetraenoic acid	C_20_H_32_O_3_	320.2351	319.2287	50.50	−ve	[C_20_H_32_O_3_]^−H^, *m/z* 275.2378, *m/z* 273.2217, *m/z* 205.1217, *m/z* 153.1275
72	Ginkgolic acid	C_22_H_34_O_3_	346.2508	347.2590	51.83	+ve	[C_22_H_34_O_3_]^+H^, *m/z* 329.2486, *m/z* 233.1530, *m/z* 189.0919, *m/z* 161.0603, *m/z* 133.0294
				345.2442	51.73	−ve	[C_22_H_34_O_3_]^−H^, *m/z* 301.2531, *m/z* 299.2372, *m/z* 203.1433, *m/z* 175.1123, *m/z* 133.0651
73	N-Isobutyl-(2E,4E)-octadecadienamide	C_22_H_41_NO	335.3188	336.3278	54.54	+ve	[C_22_H_41_NO]^+H^, *m/z* 322.3121, *m/z* 280.2647, *m/z* 182.1551, *m/z* 154.1237, *m/z* 135.1176
74	Pipnoohine	C_24_H_43_NO	361.3345	362.3438	55.42	+ve	[C_24_H_43_NO]^+H^, *m/z* 348.3279, *m/z* 306.2809, *m/z* 264.2334, *m/z* 191.1805, *m/z* 154.1238, *m/z* 135.1178

**Table 3 pharmaceuticals-14-00297-t003:** Validation parameters for marker components in Divya-Swasari-Vati (DSV) (batch #B SWV117) using HPLC–DAD analysis.

Parameters	Acceptance Criteria	Results Obtained										
		Gallic Acid	Protocatechuic Acid	Methyl Gallate	Ellagic Acid	Coumarin	Cinnamic Acid	Glycyrrhizin	Eugenol	6-Gingerol	Piperine	Glabridin
Specificity	No interference at retention time	In compliance										
Linearity	Correlation coefficient (*r*^2^) NLT 0.99	0.9992	0.9991	0.9992	0.9992	0.9982	0.9995	0.9974	0.9972	0.9975	0.9974	0.9992
	Range (μg/g)	20.0–2000	20.0–2000	6.6–2000	20.0–2000	6.6–2000	3.0–2000	20.0–2000	20.0–2000	20.0–2000	6.6–2000	6.6–2000
Precision												
Intraday	%RSD NMT 2	1.13	0.32	0.34	0.67	0.96	0.49	1.55	1.16	0.13	0.86	0.93
Interday	%RSD NMT 2	1.08	0.44	1.36	1.01	1.52	0.17	0.47	1.72	0.39	1.75	0.68
Mean average recovery (%)	90–110%	96.12	95.29	93.60	94.65	95.30	95.43	97.40	97.54	94.47	92.75	100.13
Ruggedness	NMT 10	1.13	1.91	2.79	3.26	3.94	6.92	3.79	2.05	6.87	4.20	5.22
Robustness												
Flow rate	%RSD NMT 20	2.66	9.56	15.63	6.41	5.26	6.86	7.80	2.13	4.65	2.70	7.48
Column temperature	%RSD NMT 20	5.51	9.61	15.15	4.09	5.18	3.23	3.74	1.72	4.05	5.60	8.47
Limit of Detection (LOD)	%RSD of area NMT 33	1.53	1.51	0.51	1.42	0.49	0.76	3.35	0.81	6.11	0.98	0.68
	LOD (μg/g)	0.33	0.33	0.11	0.33	0.11	0.05	0.33	0.33	0.33	0.11	0.11
Limit of Quantification (LOQ)	%RSD of area NMT 10	0.60	0.93	1.10	1.48	0.99	1.64	1.02	0.52	0.38	1.28	0.48
	LOQ (μg/g)	1.0	1.0	0.33	1.0	0.33	0.15	1.0	1.0	1.0	0.33	0.33

Note: All the parameters are validated as per the ICH-Q2 (R1) guidelines. NMT: Not More Than; NLT: Not Less Than.

## Data Availability

The data presented in this study are available within the article, the associated Appendix A, or on request from the corresponding author.

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
