# Peer review of "Comprehensive and Rapid Quality Evaluation Method for the Ayurvedic Medicine Divya-Swasari-Vati Using Two Analytical Techniques: UPLC/QToF MS and HPLC–DAD"

_pharmaceuticals, 2021, doi:10.3390/ph14040297_

Round 1
Reviewer 1 Report
The manuscript presented by the authors deals with the development and validation of an analytical method for Traditional Ayurvedic Preparation Divya-Swasari-Vati (DSV) in the context of its use for the management of airway diseases given the anti-inflammatory properties of several of the components that are part of this preparation.
Undoubtedly, one of the most critical challenges in the Pharmaceutical industry is developing reliable analytical methods that meet the health authorities' requirements to guarantee the population that the products offered contain the active principles in the amounts mentioned on the label. This situation is complicated in herbal products since various conditions can lead to variations in the quantities of the principles responsible for the therapeutic activity or even lead to toxicity by any other component. Thus, the authors' work is valuable since the method developed is robust and allows to reach the potential standardization of the products of the Traditional Ayurvedic Preparation Divya-Swasari-Vati.
The manuscript is well presented, and the analytical procedure is well documented, so I recommend its acceptance in Pharmaceuticals. The only minor concern is that the authors could prove that the choice of markers (line 74 and following) could prove by mentioning the appropriate references. Although information may appear to be known to a specialized public, the reviewer considers it necessary to justify these markers' choices.
Author Response
Point 1. The manuscript is well presented, and the analytical procedure is well documented, so I recommend its acceptance in Pharmaceuticals. The only minor concern is that the authors could prove that the choice of markers (line 74 and following) could prove by mentioning the appropriate references. Although information may appear to be known to a specialized public, the reviewer considers it necessary to justify these markers' choices.
Author Response:
Dear Reviewer, authors really appreciate the time and efforts put by the eminent reviewer in the reviewing our article. We are very happy to see the positive comments by the potential reviewer. The selection criteria was done on the basis of the established therapeutic activity of the markers and their availability. As mentioned in line number 273, 274. “The strategies behind the selection of targeted eleven markers were based on their availability, therapeutic activity and abundancy in a particular medicinal plant component.” Moreover, as per your valuable suggestion we have incorporated suitable references numbered 30, 31, 10, 32, 33, 34 and 35 which further justifies the marker selection criteria.
Following are the references which have been incorporated in the manuscript:
[30] Pastorino G, Cornara L, Soares S, Rodrigues F, O.M. Liquorice (Glycyrrhiza glabra): A phytochemical and pharmacological review. Phyther. Res. 2018, 32, 2329–39.
[31] Cortés-Rojas DF, de Souza CR, O.W. Clove (Syzygium aromaticum): a precious spice. Asian Pac. J. Trop. Biomed. 2014, 4, 90–96.
[10] Priyashree, S.; Jha, S.; Pattanayak, S.P. Bronchodilatory and mast cell stabilising activity of Cressa cretica L.: Evaluation through in vivo and in vitro experimental models. Asian Pac. J. Trop. Med. 2012, 5, 180–186, doi:10.1016/S1995-7645(12)60021-2.
[32] Jawhari FZ, Moussaoui AE, Bourhia M, Imtara H, Saghrouchni H, Ammor K, Ouassou H, Elamine Y, Ullah R, Ezzeldin E, M.G. Anacyclus pyrethrum var. pyrethrum (L.) and Anacyclus pyrethrum var. depressus (Ball) Maire: Correlation between Total Phenolic and Flavonoid Contents with Antioxidant and Antimicrobial Activities of Chemically Characterized Extracts. Plants. Plants 2021, 10, 149
[33] Rai, S.; Mukherjee, K.; Mal, M.; Wahile, A.; Saha, B.P.; Mukherjee, P.K. Determination of 6-gingerol in ginger (Zingiber officinale) using high-performance thin-layer chromatography. J. Sep. Sci. 2006, 29, 2292–2295, doi:10.1002/jssc.200600117
[34] Zahoor, M.; Zafar, R.; Rahman, N.U. Isolation and identification of phenolic antioxidants from Pistacia integerrima gall and their anticholine esterase activities. Heliyon 2018, 4, e01007, doi:10.1016/j.heliyon.2018.e01007.
[35] Sulekha, G.; Tambe Esha Identification of Chemical Constituents of Cinnamon Bark Oil by GCMS and Comparative Study Garnered from Five Different Countries. Glob. J. Sci. Front. Res. C Biol. Sci. 2019, 19, 35–42.
Reviewer 2 Report
The paper describes the development of a simple, reliable, and sensitive method for quality control of Divya-Swasari-Vati(DSV) formulation by using two analytical techniques UPLC/QTof-MS and HPLC-DAD. This last method was validated following the ICH guideline. The bibliography is actual and appropriate. The methodology used, especially those used for identification of signature markers present in the product is novel for this type of compounds and matrices, although the sample extraction and preparation procedures are, relatively, complex.
It is generally well written with few errors and too long (for example, figure 2 could deleted since that information is included in Table 2). The purpose of the study is well explained in the introduction and appears to be adequately referenced. The conclusions are brief but summarize the results well.
However, in my opinion, the paper could improve in some aspects, for example, by using the Design of experiments (DoE) strategy in the optimization and validation process (Robustness assessment). In addition, the aim of this study should show this fact. In this point, the authors used the old “OVAT” (one-variable at time) strategy in the optimization process or the authors called “trial and error procedures (see line 313). All this could increase the quality and interest to the readers.
In opinion of this reviewer, the manuscript is not suitable for publication in the Pharmaceuticals Journal in the actual form.
General comments
-Lines 313-330: The following parameters were iteratively evaluated according to an “trial and error procedures: mobile phase composition, flow, λ and chromatographic column. In addition, injection volume and column temperature were also analysed (see lines 439-440). This type of study is a good example to apply the DoE approach.
Other example of this can found in the instrumentation and chromatographic conditions: the authors used a mobile phase with two or three components, column temperature and flow rate. What is the possible effect of these factors about the response? The DoE approach could answer to this question.
-Section: 4.3 Method validation. The trueness and precision were assessed carrying out three independent determinations on days different (at least, two days) at three concentration level. This is Ok. The goal is to determine the inter-and-intra-days variability. Why the authors didn´t use this same strategy to analyse the linearity?
The linearity (lines 446-449) of the calibration curves should be evaluated by linear regression using, at least, six calibration points within the calibration range How many concentrations were used for each compound? In addition, the authors should check the null hypothesis of the lack of fit to confirm that the data follows a linear relation in the calibration range; the variability between days was not analysed. Usually, the linearity must be evaluated in different days in order to know the variability inter-and intra- days. It is important to check that there is no difference between days in order to pool data. The statistical analysis is incomplete, for example, the residual sum of squares (RSS) should be provided to know the total error. In my opinion, it is not sufficient with the correlation coefficient to say that a good linearity (or fit) was achieved (see line 348).
- Lines 244-251Robustness and Ruggedness
I am agreed with the sentence “the robustness of analytical method should show the reliability with respect to deliberate variations in the method performance”. However, the authors did not follow this criterion, any information about deliberate variations was provided or very poor (Table 3). Other question “it was observed that %RSD was within the limit of not more than 20% indicating the robustness of the method (line 246-247). I don’t understand this sentence. Why 20%? when the precision was lower than 2% for the all compound, even below 1% for many of them (Table 3).
-Lines 82-83 “the validated method was then successfully applied for the content uniformity analysis of target compounds in five batches of DSV” and lines 284-286 comments this issues again, but any results on the application of the content uniformity test was provided. The authors provided only data about the simultaneous estimation of the targeted analytes in five different batches (see lines 367-374).
Other minor comments
-Figure 4 should delete the chemical structure in order to facilitate the data visual inspection.
Author Response
Dear Reviewer, thank you very much for the critical reading of the manuscript and the comments as well as suggestions to improve the quality of this manuscript. We greatly appreciate the efforts for handling and reviewing our manuscript as well as the valuable comments. Respective revisions that provide significant help for the manuscript has been made. Please find our responses to the esteemed reviewer point by point.
Point 1. It is generally well written with few errors and too long (for example, figure 2 could deleted since that information is included in Table 2).
Response: Thank you very much for the insights. Yes indeed Figure 2 was sort of obsolete in the overall scheme of things. This figure was included to demonstrate the mass fragmentation of selected markers used in HPLC-DAD analysis. It has now been moved to the supplementary section of the manuscript.
Point 2. Lines 313-330: The following parameters were iteratively evaluated according to a “trial and error procedures: mobile phase composition, flow, λ and chromatographic column. In addition, injection volume and column temperature were also analysed (see lines 439-440). This type of study is a good example to apply the DoE approach.
Other example of this can found in the instrumentation and chromatographic conditions: the authors used a mobile phase with two or three components, column temperature and flow rate. What is the possible effect of these factors about the response? The DoE approach could answer to this question.
Response: We appreciate esteemed reviewer for this thought-provoking query/comment. Yes, we have opted OVAT (one variable at time) approach, we achieved the optimum resolution of the targeted analytes by performing 3 to 4 variations in the terms of the mobile phase, column temperature and flow rate. Thus we did not opt for the design of experiment approach at this time, as we wanted to give more emphasis on validation of the method. Our findings confirmed that the developed method can be used for the routine analysis of the formulation. However, we would now utilize the DOE approach, in our upcoming experimental work; and would like to observe the correlations. Thank you once again for these wonderful insights, it would add more quality parameters in our analytical chest of tools.
Point 3. Section: 4.3 Method validation. The trueness and precision were assessed carrying out three independent determinations on days different (at least, two days) at three concentration level. This is Ok. The goal is to determine the inter-and-intra-days variability. Why the author’s didn´t use this same strategy to analyse the linearity?
Response: We would like to thank you very much for review and all pertinent comments on our manuscript. In response to this we would like to explain that, as per the recommendation of ICH Q2(R1) the linearity is evaluated across the range of the analytical procedure. There is no such specifications of performing the linearity on inter and intra days. Because the parameter remains constant if analysed on the same or different day. Moreover, inter-and intra- days variabilities were assessed using precision studies by analyzing the actual constituent present in the formulation and by calculating the standard deviation between inter-and intra- day results. Thus, with the adherence to the ICH Q2(R1) guidelines we assessed the linearity on the same day itself.
Point 4. The linearity (lines 446-449) of the calibration curves should be evaluated by linear regression using, at least, six calibration points within the calibration range. How many concentrations were used for each compound? The linearity (lines 446-449) of the calibration curves should be evaluated by linear regression using, at least, six calibration points within the calibration range. How many concentrations were used for each compound? In addition, the authors should check the null hypothesis of the lack of fit to confirm that the data follows a linear relation in the calibration range; the variability between days was not analysed. Usually, the linearity must be evaluated in different days in order to know the variability inter-and intra- days. It is important to check that there is no difference between days in order to pool data. The statistical analysis is incomplete, for example, the residual sum of squares (RSS) should be provided to know the total error. In my opinion, it is not sufficient with the correlation coefficient to say that a good linearity (or fit) was achieved (see line 348).
Response: Dear Reviewer, thank you very much for this comment. In response to this suggestion we would like to explain that as per the recommendations of ICH Q2(R1) for the establishment of linearity, a minimum of 5 concentrations are recommended. In our study, for gallic acid, protocatechuic acid, ellagic acid, glycyrrhizin, eugenol and 6-gingerol, linearity was assessed using 7 calibration points (1 µg/ml-100 µg/ml). For, methyl gallate and glabridin, 8 calibration points were taken into consideration (0.33 µg/ml-100 µg/ml). For piperine and coumarin 7 calibration points were considered (0.33 µg/ml-100 µg/ml). Whereas, for cinnamic acid, 8 calibration points were used for linearity (0.15 µg/ml- 100 µg/ml). The lower point of linearity demonstrates the LOQ of the particular analyte. Our aim was to identify the marker components in the formulation using UPLC/QToF-MS and develop a simple and novel methodology for the simultaneous estimation of the selected analytes that can be used for the routine analysis of the product using HPLC-DAD analysis and demonstrate that the method can be applied for the batch analysis. In accordance to your suggestion we have performed residual sum of square analysis (RSS) on the individual targeted analytes. The RSS and sum of square values (SS) have been provided in Table S1 of the supplementary section. The smaller RSS values in comparison to the SS further justified the linearity. Authors, would like to thank the reviewer once again for his valuable insight.
Point 5. - Lines 244-251Robustness and Ruggedness
I am agreed with the sentence “the robustness of analytical method should show the reliability with respect to deliberate variations in the method performance”. However, the authors did not follow this criterion, any information about deliberate variations was provided or very poor (Table 3). Other question “it was observed that %RSD was within the limit of not more than 20% indicating the robustness of the method (line 246-247). I don’t understand this sentence. Why 20%? when the precision was lower than 2% for the all compound, even below 1% for many of them (Table 3).
Response: Dear Reviewer, thank you very much for this comment. In response to this suggestion we would like to explain that, In case of robustness a deliberate variations in the terms of flow rate and column temperature were taken into the consideration, results showed that % RSD of methyl gallate was found to be more than 15 % whereas for the rest of the targeted analytes the % RSD was less than 10 %. In our study, as shown in the robustness section of Table 3, we calculated the % RSD for robustness with 18 determinations including 6 initial values and 12 values from the deliberated changes (temperature ± 2 °C and flow rate ± 0.5 ml/min) respectively. The limit assigned 20 % is for the 18 determinations as described above and not for the 6 determinations individually for the respective deliberate change. Hence we kept the prescribed limit NMT 20% for the robustness parameter. Thus, if someone wants to analyze methyl gallate in similar type of product then strict adherence to the proposed column temperature and flow rate is required. Hence we recommend not to change the flow rate and column temperature while performing the analysis.
Point 6. -Lines 82-83 “the validated method was then successfully applied for the content uniformity analysis of target compounds in five batches of DSV” and lines 284-286 comments this issues again, but any results on the application of the content uniformity test was provided. The authors provided only data about the simultaneous estimation of the targeted analytes in five different batches (see lines 367-374).
Response: Dear Reviewer, we are very grateful for this very important comment. We would like to sincerely apologize for our typo mistake which says the “validated method was then successfully applied for the content uniformity analysis of target compounds in five batches of DSV”. The terminology content uniformity has been removed. Author wished to write simultaneous detection of the targeted analytes which has been rectified now. The prime aim of our research was to develop a novel method which can be used for the routine analysis of the Divya-Swasari-Vati formulation. The HPLC-DAD analysis simultaneously quantified the targeted analytes in 5 different batches of DSV formulation. We have represented the data in the form of mean and SEM for the individual targeted analytes which are mentioned in line number 361 and 362.
Point 7. Figure 4 should delete the chemical structure in order to facilitate the data visual inspection.
Response: Structures have been moved off from the graph, as per your recommendation for facilitation of visual data inspection.
Reviewer 3 Report
The manuscript Comprehensive and Rapid Quality Evaluation Method of traditional Ayurvedic preparation Divya-Swasari-Vati using two analytical techniques UPLC/QToF/MS and HPLC-DAD fits the scope of the journal. The authors present a simple method of identification and quatification and its validation for some phytochemicals from an herbal preparation. The results, discussion and materials and methods are somewhat mixed and should be carefully separated.
Before publication, I would like to ask the authors to clarify the following:
Figure 1 The quality of the chromatograms is poor - please replace them
Figure 2 – should be included in supplementary material
2.3 Validation of the Developed and Optimized HPLC Method for Quantitative Analysis of Eleven Marker compo-nents in DSV – the section is a mixture of results and method (part which is partially described in the section Materials and method); please refer only to the results in this section.
Rows 312-330, 337-366 – these paragraphs should also be included in the Materials and method section of the article; please specify the literature which stood at the base of the selection of the mobile phases and other chromatographic condition.
Rows 367-377 – are the samples different? What are the differences (are statistical significant)? Are any regulations / limits respected or not? What was the purpose of analyzing 5 different batches?
377-378 – the quality of a supplement can be assessed depending on each specific regulations. Thus, the utilization of a method who can identify and assay several phytochemicals can be easily replaced by other instrumental method such as: FTIR, NIR, HPLC, HPTLC –fingerprinting – these methods can be used to assess the uniformity of the batches and for the detection of some main components. The authors should extend the discussions on the importance of the other components (the Ca sources - table 1 components 10-16). Being a mixture powder containing metal oxides and other components, these components can greatly affect the quantitative analysis of phytochemicals.
- Materials and Methods
Sample preparation
How sample preparation was established? Components 1-16 (table 1) influence the solubility of the components? Were any other sample preparation methods tested?
UPLC/QToF MS Analysis – Please add the reference for the method which was the basis of development of the current protocol.
Author Response
Author Response:
Dear Reviewer, We would like to thank you very much for preparing this review and all pertinent comments on our manuscript. We greatly appreciate the efforts for handling and reviewing our manuscript as well as the valuable comments. Respective revisions that provide significant help for the manuscript have been made. Please find our response to the queries point by point below.
Point 1. Figure 1 the quality of the chromatograms is poor - please replace them
Response: Dear Reviewer, thank you very much for your suggestion we have replaced the image with the high resolution one.
Point 2. Figure 2 – should be included in supplementary material
Reply: Figure number two has been included in the supplementary material.
Point 3. Validation of the Developed and Optimized HPLC Method for Quantitative Analysis of Eleven Marker components in DSV – the section is a mixture of results and method (part which is partially described in the section Materials and method); please refer only to the results in this section.
Response: Dear Reviewer, thank you very much for this comment. We agree with this suggestion, and have segregated the results and methods section, in the manuscript.
Point 4. Rows 312-330, 337-366 – these paragraphs should also be included in the Materials and method section of the article; please specify the literature which stood at the base of the selection of the mobile phases and other chromatographic condition.
Response: Dear Reviewer thank you very much for this comment. As per your suggestion the paragraphs of line 312-330, 337-366 have also been included in the material and methods section. The method is novel, which was developed based on analysts own experience and trial and error methodology hence the analyst didn’t referred any particular literature as such. But, have referred the pharmacopoeial and FDA requirements for the method development in order to achieve the requirements like resolution and specificity of the targeted analytes. The appropriate references has been added to the manuscript, numbered [43 and 44]
Incorporated references are:
[43] U.S. Department of Health and Human Services, F. and D.A. Bioanalytical Method Validation Guidance for Industry. U.S. Dep. Heal. Hum. Serv. Food Drug Adm. 2018.
[44]. USP (1225) Validation of Compendial Procedures. United States Pharmacopoeia XXXVII Natl. Formul. XXXII 2007.
Point 5. Rows 367-377 – are the samples different? What are the differences (are statistical significant)? Are any regulations / limits respected or not? What was the purpose of analyzing 5 different batches?
Response: Yes, these are different samples obtained from 5 different formulation batches of Divya- Swasari-Vati. It is a commercial batch, manufactured under GMP compliance facility. There are no such regulatory requirements in terms of limit for the selected compounds. The purpose of analyzing 5 different batches was to confirm that the developed technique is pertinent for the simultaneous detection of the targeted marker analytes in considered batches, which is applicable for the routine quality control analysis of the formulation.
Point 6. 377-378 – the quality of a supplement can be assessed depending on each specific regulations. Thus, the utilization of a method who can identify and assay several phytochemicals can be easily replaced by other instrumental method such as: FTIR, NIR, HPLC, HPTLC –fingerprinting – these methods can be used to assess the uniformity of the batches and for the detection of some main components. The authors should extend the discussions on the importance of the other components (the Ca sources - table 1 components 10-16). Being a mixture powder containing metal oxides and other components, these components can greatly affect the quantitative analysis of phytochemicals.
Response: Dear Reviewer, thank you very much for this comment. For this we would again like to explain that our research proposes the development of a simple, reliable, and sensitive method for quality control of Divya-Swasari-Vati (DSV) formulation. The HPLC-DAD analysis simultaneously quantified the targeted analytes in 5 different batches of DSV formulation is applicable for the routine control analysis of the formulation. As per your suggestions we have included the importance of bhasmas in the discussion part. We agree with the fact that the formulation also contains calcio-mineral components such as Bhasmas. The solvent used for the sample preparation consisted of methanol: water (80:20) i.e. 80% methanol, and the bhasmas which are inorganic in nature are insoluble in the considered solvent media. Moreover they also lack the U.V and visible absorptivity, thus they will not interfere during the DAD analysis of the phytochemicals in the formulation.
Point 7. How sample preparation was established? Components 1-16 (table 1) influence the solubility of the components? Were any other sample preparation methods tested?
Response: Different combination of solvent system were tried like methanol: water (20:80), (50:50), and (80:20) to achieve the maximum sample solubility. Out which the maximum solubility was achieved using methanol: water (80:20). We agree with the fact that the formulation also contains calcio-mineral components such as Bhasmas. The solvent used for the sample preparation consisted of methanol: water (80:20), and the bhasmas which are inorganic in nature are insoluble in the considered solvent media (methanol), which will not affect the solubility of the phytochemicals. Moreover, we are calculating the content in % w/w, so insoluble matrices present in the solution are considered in reporting the result.
Point 8. UPLC/QToF MS Analysis – Please add the reference for the method which was the basis of development of the current protocol.
Response: The novel UPLC/QToF MS method was developed based on our in-house protocol and analysts own expertise for the identification of 74 metabolites in the formulation. As per your valuable suggestion appropriate reference which assisted in the method development has been added in the manuscript.
The following reference has been added in the manuscript:
[30].Vogeser, M.; Seger, C. A decade of HPLC–MS/MS in the routine clinical laboratory — Goals for further developments e. Clin. Biochem. 2008, 41, 649–62.
Round 2
Reviewer 2 Report
The authors have addressed my questions and suggestions. I believe the manuscript in the actual form can published in Pharmaceuticals.
Reviewer 3 Report
The authors made the corrections and they clarified all issues raised. The manuscript has been improved and therefore I consider that it is suitable for publication in Pharmaceuticals.